# Fracture Performance Study of Carbon-Fiber-Reinforced Resin Matrix Composite Winding Layers under UV Aging Effect

**DOI:** 10.3390/ma17040846

**Published:** 2024-02-09

**Authors:** Zhen Liu, Feiyu Zhou, Chao Zou, Jianping Zhao

**Affiliations:** 1School of Mechanical and Power Engineering, Nanjing Tech University, Nanjing 211816, China; liuzhen_929@163.com (Z.L.);; 2Jiangsu Key Lab of Design and Manufacture of Extreme Pressure Equipment, Nanjing 211816, China

**Keywords:** CFRP laminates, UV aging, fracture toughness, extended finite element, crack analysis, type IV hydrogen storage cylinders

## Abstract

There is limited research on the fracture toughness of carbon-fiber-reinforced polymer (CFRP) materials under accelerated UV aging conditions. In this study, the primary focus was on investigating the influence of varying durations of ultraviolet (UV) irradiation at different temperatures on the Mode I, Mode II, and mixed-mode fracture toughness of CFRP laminates. The results indicate that with increasing UV aging duration, the material’s Mode I fracture toughness increases, while Mode II fracture toughness significantly decreases. The mixed-mode fracture toughness exhibits an initial increase followed by a subsequent decrease. Furthermore, as the aging temperature increases, the change in the fracture toughness of the material is more obvious and the rate of change is faster. In addition, the crack expansion of the composite layer of crack-containing Type IV hydrogen storage cylinders was analyzed based on the extended finite element method in conjunction with the performance data after UV aging. The results reveal that cracks in the aged composite material winding layers become more sensitive, with lower initiation loads and longer crack propagation lengths under the same load. UV aging diminishes the overall load-bearing capacity and crack resistance of the hydrogen storage cylinder, posing increased safety risks during its operational service.

## 1. Introduction

Carbon Fiber Reinforced Polymer (CFRP) is a high-performance composite material composed of carbon fibers and an epoxy resin matrix. Its unique characteristics of lightweight, high strength, and high rigidity make it play a crucial role in various fields such as aerospace, rail transportation, pressure vessels, and civil engineering [1,2]. In the realm of pressure vessels, Type IV hydrogen storage cylinders have garnered international attention due to their advantages of high pressure, large capacity, high hydrogen storage density, and lightweight construction [3,4]. However, during prolonged service, CFRP undergoes aging due to factors such as ultraviolet radiation, humidity, temperature, and mechanical loading, leading to material degradation and a reduction in the load-bearing capacity of the cylinders [5]. Therefore, in the design process, it is imperative to better understand and consider the impact of environmental factors such as humidity, temperature, and ultraviolet radiation on the durability, reliability, and safety of Type IV hydrogen storage cylinder CFRP structures.

Typically, polymer materials undergo simultaneous physical aging and chemical aging under conditions of high temperature and UV exposure. Prolonged exposure to temperatures below the glass transition temperature (Tg) results in physical aging, characterized by the slow evolution of the polymer towards thermodynamic equilibrium, leading to a partial enhancement of certain mechanical properties during this process [6]. On the other hand, chemical aging primarily manifests as resin material photooxidation, thermal oxidation, and photolysis reactions induced by ultraviolet radiation [7]. This results in the degradation of unsaturated polyester and vinyl ester resins in CFRP, with molecular chain and aromatic chemical bond breakage in the resin [8,9,10], potentially affecting the material’s appearance and performance. Ahmad G.K. et al. [11], using various characterization tools such as transmission electron microscopy, colorimeter, and optical microscope, confirmed the appearance of microcracks and discoloration on the surface of epoxy coatings after 1000 h of UV exposure. Shi, Z. et al.’s study results [12] indicate that UV aging has a deteriorating effect on multiple mechanical properties of CFRP. After 80 days of UV exposure, significant reductions in macroscopic mechanical performance, particularly in longitudinal compressive strength, were observed. Both strength and modulus decreased with increasing UV irradiation time. In the early stages of UV aging, the crosslinking of molecular chains due to physical aging led to some enhancement of mechanical properties, while in the later stages, a decline in these properties occurred.

In the study of fracture toughness of CFRP materials, the interlaminar strength of CFRP is significantly lower than the in-plane tensile and compressive strength, making it prone to delamination failure. Additionally, UV aging has a destructive effect on both the matrix and the interlaminar interface. Therefore, investigating the interlaminar fracture toughness of CFRP materials under the influence of UV aging is crucial. Ashrafi et al. [13] observed the degradation behavior of polymer and fiber/matrix interfaces under UV radiation conditions using scanning electron microscopy (SEM). The results showed the occurrence of microcracks in the matrix under UV radiation and condensation conditions. The presence of these microcracks directly affects the interlaminar fracture toughness of CFRP materials [14]. Currently, the fracture toughness testing of composite laminate includes Mode I interlaminar fracture toughness, Mode II interlaminar fracture toughness, and mixed-mode interlaminar fracture toughness testing. Hamed F. [15], through DCB and ENF tests measuring Mode I and Mode II fracture toughness, found that a stronger fiber-matrix interface significantly improves the Mode I and Mode II fracture toughness of CFRP materials. Liu, JM et al. [16] used polyetherimide (PEI) and polyetheretherketone (PEEK) as an intermediate layer in composite materials co-curing connections, and tested their mixed-mode fracture toughness. Scarselli, G et al. [17] evaluated the Mode I fracture toughness of untreated, UV-aged, and plasma-treated glass fiber-reinforced thermoplastic plates. The results indicated an overall reduction in the Mode I fracture toughness of UV-aged and plasma-treated glass fiber-reinforced thermoplastic plates.

ABAQUS mainly employs four techniques for simulating crack propagation, including Extended Finite Element Method (XFEM), cohesive model, debond, and collapse element [18]. Compared to other methods, XFEM does not require a predefined crack propagation path, allowing the simulation of cracks in any direction with crack tips located at arbitrary positions, including inside elements [19]. It also eliminates the need for extremely refined meshing in the crack-tip region [20], making it suitable for various complex models. Tasavori, M [21] used XFEM to investigate the fracture behavior of cracks on the inner surface of axisymmetric pressure vessels, calculating and comparing stress intensity factors for single-layer aluminum and outer-layer composite materials. Park, WR [22] utilized layer modeling and XFEM to assess crack behavior in Type III high-pressure hydrogen storage cylinders, with failure criteria determined based on maximum principal stress exceeding the allowable tensile strength and displacement. The results revealed a significantly higher risk of transverse damage in the composite layers of Type III high-pressure hydrogen storage cylinders compared to the fiber direction.

Currently, there is considerable research on the changes in mechanical properties and damage mechanisms of composite materials under ultraviolet (UV) radiation. However, few studies have focused on the fracture toughness of carbon fiber reinforced polymer (CFRP) materials under accelerated UV aging conditions. In this study, UV-accelerated aging experiments were conducted, and fracture mechanics performance tests were performed to investigate the influence of different durations of UV irradiation at various temperatures on the Mode I, Mode II, and mixed-mode fracture toughness of CFRP laminates. Based on the analytical model of the Arrhenius formula, these results can be further used to predict the retention of fracture toughness of CFRP materials under long-term aging [23]. Concurrently, considering the performance data before and after UV aging, an analysis of crack propagation behavior and stress conditions in Type IV hydrogen storage cylinders with cracks under high internal pressure was conducted using the Extended Finite Element Method (XFEM). These results may have significant value in ensuring the safety of Type IV hydrogen storage cylinders.

## 2. Experiment

### 2.1. Experimental Materials and Sample Preparation

The carbon fiber/epoxy resin composite materials investigated in this study were fabricated using a hot-press molding process. The prepreg for laminate preparation was sourced from Shandong Jiangshan Fiber Technology Co., Ltd. (Dezhou, China). The fibers used were T700S carbon fibers from Toray Industries, Tokyo, Japan, with an internal code of Y04. The resin employed was bisphenol A epoxy resin, and the volume fraction of carbon fibers was 60%. The fiber and resin parameters are presented in Table 1 and Table 2, respectively. Dicyandiamide was used as the curing agent, mixed with the resin at a ratio of 10:1, and the curing temperature for the resin system ranged from 90 to 130 °C.

The CFRP material used in the experiment was fabricated using a hot-press molding process, with the fiber orientation set as a unidirectional 0° layup. The final laminated panel had dimensions of 1000 × 1000 mm^2^, as shown in Figure 1. The hot-press molding process, illustrated in Figure 2, commenced with the design of the layup for the laminated panel. An industrial automated cutting machine was employed to cut the prepreg into the desired shapes. Subsequently, the layup was performed, eliminating any entrapped gases, and the assembly was vacuum-sealed and cured in a high-pressure vessel. The curing conditions for the high-pressure vessel included a temperature of 130 °C, curing time of 2 h, and pressure of 1.5 MPa, as depicted in Figure 3. Following the curing of the CFRP composite material, the system was allowed to cool down to the safe operational range for temperature, pressure, and other parameters before stopping the process. Next, the burrs of the composite were removed and numbered for backup; finally, the desired specimen was cut.

In accordance with ASTM D5528 [25], ASTM D7905 [26], and ASTM D6671 [27] standards, laminated panel specimens for Mode I (a), Mode II (b), and mixed-mode (c) interlaminar fracture were prepared. A 10 μm thick polytetrafluoroethylene (PTFE) film was embedded in the mid-plane of the laminated panel to create initial pre-existing cracks. The specimen dimensions are depicted in Figure 4.

### 2.2. UV Aging Experiment

Following the ISO 4892-2016 [28] standard, an artificial indoor UV-irradiation-accelerated aging method was employed to age the CFRP laminated panel specimens in a UV aging test chamber. The laboratory light source UVA-340 was selected to simulate global solar UV radiation. Three different blackboard temperatures (70 °C, 60 °C, and 50 °C) were set for aging, and the aging periods were chosen as 10 days, 20 days, 30 days, and 50 days. The aging process consisted of cycles with a duration of 12 h each, comprising 8 h of dry UV irradiation and 4 h of condensation stage (without UV irradiation). The UV aging tests were conducted using the KW-UV3-A UV aging test chamber from KOWINTEST (Dongguan, China), as shown in Figure 5. The aging chamber was equipped with eight fluorescent UV lamps with a wavelength of 340 nm, each with a power of 40 W, and the irradiance of UV light was set to 1 W/(m^2^·nm). The UV irradiation aging conditions are detailed in Table 3.

### 2.3. Interlaminar Fracture Performance Testing

The specimens used for interlaminar fracture performance testing are illustrated in Figure 4. Following the UV aging experiments, the specimens were retrieved, and white solvent-based coating was applied to the sides of the specimens, as depicted in Figure 6, both for the aged and unaged specimens. The specimens were then marked for easy identification and measurement of crack tips during testing. The interlaminar fracture performance testing was conducted using an INSTRON 5566 Universal Testing Machine from Instron & Co., (Shanghai, China). Throughout the experimental process, five valid samples were tested for each aging condition.

#### 2.3.1. Mode I Interlaminar Fracture Test

Mode I interlaminar fracture toughness was evaluated through a Double Cantilever Beam (DCB) test in accordance with ASTM D5528 [25]. The experimental setup is depicted in Figure 6a, with a loading rate of 2.5 mm/min and an unloading rate of 25 mm/min. Throughout the initiation and propagation of the crack, it was essential to concurrently record the load, crack opening displacement, and crack length. The specimen was loaded a second time until the total crack extension reached approximately 50 mm, after which unloading was performed.

Following the Modified Beam Theory (MBT), the calculation formula for Mode I interlaminar fracture energy release rate, $G_{IC}$, is given by:(1)$$G_{IC}=\frac{3p\delta }{2b(a+\left|\Delta \right|)}\times \frac{F}{N}$$
where: *P* is the applied load, N; δ is load line displacement, mm; *b* is specimen width, mm; ($a+\left|\Delta \right|$) is the equivalent crack length of the specimen, mm; *F* is large displacement correction factor; and *N* is loading block correction factor.

#### 2.3.2. Mode II Interlaminar Fracture Test

Mode II interlaminar fracture toughness was assessed through an End Notched Flexure (ENF) test following ASTM D7905 [26]. The ENF specimens were tested using a three-point bending fixture with a span of 100 mm, as depicted in Figure 6b. The displacement loading rate for the test was set at 0.8 mm/min. The Mode II interlaminar fracture test was conducted in two phases: Non-Pre-Crack (NPC), and Pre-Crack (PC). The purpose of the dual loading was to eliminate additional effects caused by the membrane created by the pre-existing crack. The Mode II interlaminar fracture energy release rate, $G_{IIC}$, was determined using the Compliance Calibration (CC) method [19]. According to the CC method, the specimens were marked at distances of 20 mm (a_1_), 30 mm (a_0_), and 40 mm (a_2_) from the crack tip. Calibration tests were conducted for the initial crack lengths of 20 mm (a_1_) and 40 mm (a_2_). Finally, fracture tests were performed for the initial crack length of 30 mm (a_0_).

The Mode II interlaminar fracture energy release rate based on the CC method is calculated using the following equation:(2)$$G_{IIC}=\frac{3mP_{Max}^{2}a_{0}^{2}}{2B}$$
where, $P_{Max}$ is the maximum load value on the load–displacement curve, N; *B* is the width of the specimen, mm; and *m* is compliance calibration coefficients determined through linear fitting of the curve.

#### 2.3.3. Mixed-Mode Interlaminar Fracture Test

In many real-life scenarios, composite structures may experience a combination of fracture modes. The mixed-mode fracture test characterizes the specimen’s ability to resist crack propagation under the combined effects of specific Mode I and Mode II fractures. It becomes crucial to determine the mixed-mode interlaminar fracture energy release rate, $G_{C}$, in such cases [29].

The Mixed-Mode Bending Test (MMB) was conducted according to ASTM D6671 [27] to assess the interlaminar fracture toughness under different loading ratios of Mode I to Mode II. The experimental setup is illustrated in Figure 6c, with the lever arm length ‘c’ adjusted to achieve a mixed-mode ratio $G_{II}/G$ of 0.5. The half-span length ‘L’ for the MMB test was set at 50 mm, and the loading rate was maintained at 1 mm/min. To minimize geometric non-linear effects caused by lever transmission, the loading height was consistently kept slightly higher than the connection point between the lever and the specimen. The applied load on the specimen remained perpendicular throughout the loading process.

Formulas for calculating fracture toughness have been adjusted for the weight of the lever. The measured modulus values $E_{11}$, $E_{22}$, and $G_{13}$, before and after UV aging are provided in detail in Table 4. These values are then substituted into the formulas for calculating the crack length correction factor:(3)$$\chi =\sqrt{\frac{E_{11}}{11G_{13}}[3-2{\left(\frac{\Gamma }{1+\Gamma }\right)}^{2}]}$$
where, Γ is the transverse modulus correction factor, and the calculation formula is as follows:(4)$$\Gamma =1.18\frac{\sqrt{E_{11}E_{22}}}{G_{13}}$$

Substituting the calculated values of χ and Γ into the formulas for energy release rate, the Mode I energy release rate ${(G}_{I})$, Mode II energy release rate ($G_{II}$), and total strain energy release rate ($G_{C}$) can be determined:(5)$$G_{I}=\frac{12{\left[P(3c-L)+P_{g}(3c_{g}-L)\right]}^{2}}{16b^{2}h^{3}L^{2}E_{1f}}{\left(a+\chi h\right)}^{2}$$
(6)$$G_{II}=\frac{9{\left[P(c+L)+P_{g}(c_{g}+L)\right]}^{2}}{16b^{2}h^{3}L^{2}E_{1f}}{\left(a+0.42\chi h\right)}^{2}$$
(7)$$G_{C}=G_{I}+G_{II}$$
where: *a* is the length of crack, mm; *b* is the specimen width in mm; *c* is the lever length of the MMB test apparatus in mm; $c_{g}$ is the lever length to the center of gravity in mm; $E_{1f}$ is the elastic modulus in the fiber direction measured during bending, Mpa; *h* is half the thickness of the specimen in mm; *L* is half the span length of the MMB test apparatus in mm; $P_{g}$ is the weight of the lever and the fixed device, N; $E_{11}$ is the longitudinal elastic modulus measured during tension, Mpa; $E_{22}$ is the transverse elastic modulus, Mpa; $G_{13}$ is the in-plane shear modulus, Mpa.

## 3. Experimental Results and Discussion

### 3.1. Analysis of Mode I Interlaminar Fracture Toughness Results

Figure 7 shows the Load–displacement curves (a) and R-curves (b) of type I fracture tests with different aging days at a constant aging temperature of 70 °C. From Figure 7a, it can be observed that the 5%MAX load value for the non-aged specimen is 33.7 N. With increasing aging time, the maximum load values exhibit a gradual upward trend. After aging for 50 days at 70 °C, the 5%MAX load value reaches 37.8 N, representing a 12.2% increase compared to the non-aged specimen. This phenomenon suggests that UV aging renders the CFRP material less susceptible to crack propagation in Mode I interlaminar fracture toughness testing, resulting in an enhancement of the specimen’s Mode I fracture toughness. The fracture energy release rate results are depicted in Figure 7b.

However, upon reaching the 5%MAX load point, crack propagation is observed, accompanied by the occurrence of fiber bridging, as depicted in Figure 8b. Investigations reveal that, with the increasing aging duration, the effectiveness of fiber bridging intensifies, leading to an augmentation of the material’s fracture toughness. Combining the observations from Figure 7b and Figure 8, it is evident that in the non-aged specimens, the phenomenon of fiber bridging during crack propagation is limited, and the energy release rate remains essentially unchanged. Conversely, as aging progresses, the fiber bridging effect intensifies, with bridged fibers enhancing the specimen’s resistance to crack propagation, resulting in a gradual increase in the energy release rate.

The origin of fiber bridging lies in the occurrence of delamination between the matrix and fibers under tensile stress, forming interlaminar cracks. When these cracks extend to adjacent fiber layers, some fibers, owing to their high tensile strength and elastic modulus, do not immediately rupture. Instead, they form bridges on either side of the crack, thereby enhancing the toughness of the composite material [30,31].

As depicted in Figure 9, the curve illustrates the variation of Mode I energy release rate ($G_{IC}$) under different aging temperatures and durations. From the graph, it is evident that, at a constant temperature, longer aging times correspond to higher energy release rates. The mean value of $G_{IC}$ for the non-aged specimens is 241.6 J/m^2^, reaching 294.9 J/m^2^ after 50 days of aging at 70 °C—an increase of 22.1%. This phenomenon can be attributed to the weakening of interlaminar interface bonding in the CFRP laminate with increasing aging time. However, the fiber bridging phenomenon on the upper and lower surfaces of the interlaminar region becomes more pronounced. Additionally, the condensation phase during UV aging leads to moisture absorption in the CFRP laminate, facilitating enhanced fiber bridging. Ultimately, this contributes to the observed increase in Mode I energy release rate. Moreover, under the same aging duration, higher aging temperatures result in a faster rate of change in Mode I energy release rate and elevate the upper limit of the increase in Mode I energy release rate.

### 3.2. Analysis of Mode II Interlaminar Fracture Toughness Results

Figure 10 illustrates the load–displacement curves from Mode II fracture tests conducted at a constant aging temperature of 70 °C for different aging durations. It can be observed that the non-aged specimen exhibits a maximum load value of 1053.3 N. As the aging time increases, there is a gradual decline in the maximum load, reaching 977.7 N after 50 days of aging, indicating a reduction of 7.2% compared to the non-aged specimen. The initial slope of the curve also exhibits a gradual decrease, representing a corresponding increase in the flexibility of the specimen with the increasing aging days. The slope, being the reciprocal of the initial stiffness, suggests that the specimen’s flexibility increases with the duration of aging. Furthermore, UV aging diminishes the bonding between fibers and resin, weakening the interlaminar interface bonding in the specimen. This reduction in bonding strength leads to an increase in the maximum opening displacement, indicating an extension of the crack length. Consequently, the material’s fracture toughness decreases, making it more susceptible to crack propagation.

The curves depicted in Figure 11 illustrate the variation of Mode II energy release rate ($G_{IIC}$) with aging time under different aging temperatures. Analysis of this graph reveals that, at a constant temperature, longer aging times correspond to lower energy release rates. Specifically, the non-aged specimen exhibits a Mode II energy release rate ($G_{IIC}$) of 849.4 J/m^2^, which decreases to 694.8 J/m^2^ after 50 days of aging at 70 °C, representing an 18.2% reduction compared to the non-aged specimen. When the aging duration is constant, higher aging temperatures result in a more pronounced and faster decline in the specimen’s energy release rate. The 50-day aging at 70 °C shows a 10.8% greater reduction compared to aging at 50 °C. This phenomenon is attributed to the UV aging-induced weakening of interlaminar bonding in CFRP, and the increasing extent of fiber-matrix debonding with prolonged aging time and elevated aging temperatures in the ENF specimens. This, in turn, affects the specimen’s resistance to crack propagation and failure, leading to a gradual decrease in Mode II energy release rate ($G_{IIC}$). Furthermore, the analysis of Figure 11 indicates a gradual decrease in the slope of the curve with increasing aging time, suggesting a noticeable slowing of the rate of decline in Mode II energy release rate. This effect becomes particularly prominent after 30 days of aging, potentially indicating that UV-induced damage to the matrix and fiber/matrix interface in the CFRP laminate reaches saturation, resulting in the energy release rate reaching its lowest point.

### 3.3. Analysis of Mixed-Mode Interlaminar Fracture Toughness Results

Figure 12 depicts the load–displacement curves from mixed-mode fracture tests conducted at a constant aging temperature of 70 °C for various aging durations. Analysis of the graph indicates that the 5% MAX load value for the non-aged specimen is 236.41 N. With increasing aging time, the maximum load exhibits an initial ascent followed by a gradual decline. After 50 days of aging, the 5% MAX load value decreases to 213.50 N, representing a reduction of 9.7% compared to the non-aged specimen.

Figure 13 presents the variation of mixed-mode energy release rate ($G_{C}$) with time at different aging temperatures. From the graph, it can be observed that the non-aged specimen exhibits a relatively low $G_{C}$ value of 347.9 J/m^2^. As the aging time increases, the $G_{C}$ value initially rises, reaching its peak at 373.9 J/m^2^ after 10 days of aging at 70 °C, followed by a subsequent decline. After 50 days of aging, the $G_{C}$ value is recorded as 355.0 J/m^2^. The analysis is as follows: During the early stages of UV aging, there is an initial increase in $G_{C}$. This is attributed to the post-curing of the resin, resulting in cross-linking of molecular chains and an enhancement in material toughness. However, as the aging time progresses, a noticeable decline is observed. At this point, the detrimental effects of UV radiation outweigh the beneficial effects of resin post-curing. The interlaminar bonding strength in CFRP significantly decreases, and UV irradiation intensifies the extent of delamination between the fibers and the matrix. This ultimately leads to a decrease in the total mixed-mode fracture toughness $G_{C}$ value.

## 4. XFEM-Based Finite Element Simulation Method

### 4.1. Wound Composite Modeler (WCM) and XFEM Technology

The composite winding layer of Type IV hydrogen storage cylinders is formed by wrapping carbon fibers impregnated with epoxy resin on the inner liner of the container. As the materials employed exhibit orthotropic anisotropy, their properties depend on the winding angles. Therefore, the most suitable approach is to employ a laminated modeling method, the principles of which are illustrated in Figure 14.

This method involves the utilization of the Wound Composite Modeler (WCM), which is an ABAQUS plugin tool designed for convenient modeling of composite pressure vessels. In conventional analytical approaches, the continuously changing shape of the dome section, fiber overlapping, and variations in thickness and angles make analysis challenging. By contrast, The WCM can calculate the changes of fiber angles over the dome, and applies it to the finite element model (FEM) [32]. The underlying principles are illustrated in Figure 15 [33]. Therefore, the WCM enables accurate analysis of the entire vessel.

One approach in Abaqus for simulating crack propagation based on the Extended Finite Element Method (XFEM) framework is rooted in cohesive zone behavior. The formulas and rules employed to control crack propagation bear a strong resemblance to those used for cohesive elements exhibiting traction-separation constitutive behavior, and are applicable to elastoplastic analyses [34]. The primary advantages of XFEM lie in its ability to eliminate the need for special treatment of the mesh around the crack tip and to provide relatively accurate predictions. This significantly streamlines the simulation process and allows for the modeling of more complex, irregular crack shapes and paths [19].

XFEM employs shape functions with discontinuity terms to represent displacements within the computational domain, and such elements are referred to as enriched elements (within which there cannot exist two separate cracks). Throughout the computation, the representation of discontinuous fields is entirely independent of the mesh boundaries [35]. When the influence region of a node is divided into two disjointed parts by a crack, the corresponding ghost node is activated. In Abaqus [36], the displacement vector around the crack in XFEM is expressed as:(8)$$u(x)=\sum_{j=1}^{n}N_{j}(x)\left[u_{j}+H(x)a_{j}+\sum_{a=1}^{ne(j)}N_{k}(x)\sum_{\alpha =1}^{4}F_{\alpha }(x)b_{j}^{\alpha }\right]$$
where, *H*(*x*) and $F_{\alpha }(x)$ represent the jump function and the crack-tip asymptotic displacement function, respectively. *N_j_*(*x*) denotes the conventional nodal shape functions, *u_j_* is the displacement vector of the continuous part of the finite element solution, *ne*(*j*) represents the total number of enrichment functions for node *j*, *a_j_* is the nodal enrichment degree of freedom vector for the entire crack surface, and $b_{j}^{\alpha }$ is the nodal enrichment degree of freedom vector for the elements associated with the crack-tip asymptotic displacement function $F_{\alpha }(x)$.

In Abaqus, there are six fracture initiation criteria for crack damage, each providing corresponding fracture initiation thresholds to assess whether crack damage has initiated. In this study, the Maximum Principal Stress Criterion will be employed as the XFEM fracture initiation criterion for investigating crack propagation in the Type IV hydrogen storage cylinder winding layer. The analysis utilizes the Maximum Principal Stress Theory to assess crack propagation, considering only the relationship between the stress in the fiber direction ($\sigma_{1}$) and the stress in the transverse direction ($\sigma_{2}$), as shown in Equation (9). If any of the expressions below is greater than or equal to the allowable strength in each direction, it is considered that the crack can propagate. The maximum principal stresses $\sigma_{p1}$ and $\sigma_{p2}$ at each location were employed as the starting criteria for damage, as follows:(9)$$\left\{\begin{array}{l} \frac{\sigma_{1}}{X_{t}}\geq 1\\ \frac{\sigma_{2}}{Y_{t}}\geq 1 \end{array}\right.$$

### 4.2. Modeling of Type IV Hydrogen Storage Cylinder with Cracks

The finite element model of the Type IV hydrogen storage cylinder comprises the BOSS (Bossed O-ring Seal System) structure, inner liner, and fiber winding layer. The material chosen for the composite winding layer is carbon fiber (T700S) with epoxy resin. The impact of UV aging on the mechanical properties of carbon fiber composite materials is extensively explored in reference [12], where the post-aging material parameters are summarized. The material parameters for CFRP obtained after aging for 50 days in a 70 °C UV aging chamber (equivalent to approximately 2 years of natural sunlight exposure) are presented in Table 4. The BOSS structure is constructed from AL6061-T6 material, and the inner liner is made of PA6 plastic, with material properties listed in Table 5.

**Table 4 materials-17-00846-t004:** Material parameters of T700S/epoxy composite.

T700S/Epoxy Composite	Non-Aging	After Aging
$E_{11}$ (Mpa)	149,200	128,250
$E_{22},E_{33}$ (Mpa)	8150	8330
$\nu_{12},\nu_{13}$	0.28	0.28
$\nu_{23}$	0.336	0.336
$G_{12},G_{13}$ (Mpa)	4260	4490
$G_{23}$ (Mpa)	3710	3710
$X_{t}$ (Mpa)	2585.11	2239
$Y_{t}$ (Mpa)	50.5	50.69
$X_{c}$ (Mpa)	1477	1158.4
$Y_{c}$ (Mpa)	180	180
S (Mpa)	72.47	74.45

The study by Park, WR [22] revealed that, due to the operational conditions of high-pressure hydrogen storage cylinders, the risk of transverse damage in the composite winding layer is significantly greater than that in the fiber direction. Under given pressure conditions, cracks perpendicular to the length direction of the vessel are more prone to causing damage than parallel cracks. Therefore, in this research, cracks oriented perpendicular to the length direction of the vessel are selected to investigate the influence of UV radiation on cross-layer crack propagation.

Using the finite element analysis software Abaqus-WCM (2017), a Type IV composite hydrogen storage cylinder with a pre-existing crack is modeled. The bottom opening of the cylinder is constrained with ENCASTRE, while the top opening adopts ZASYMM constraints, ensuring U1 = U2 = UR3 = 0. The structural configuration, dimensions, crack location, boundary conditions, and loading are depicted in Figure 16.

The Type IV cylinder is designed for a working pressure of 52 MPa with a safety factor of 2.25. The burst pressure is 117 MPa, and the total thickness of the composite material’s circumferential winding and helical winding layers is 32.542 mm. To enhance convergence in the finite element analysis, the layering scheme of the composite material is simplified, as shown in Table 6. The meshing of the composite layers is automatically performed by WCM, and relevant parameters are set to control the mesh quantity. The element types selected include four-node bilinear axisymmetric quadrilateral elements (CAX4) and three-node linear axisymmetric triangular elements (CAX3). A colored schematic of the layered material is shown in Figure 17.

### 4.3. The Influence of UV Aging on the Crack Propagation of Winding Layers

XFEM cracks are created in the Interaction module, with an initial crack length of 1.228 mm, located in the second layer of the composite material winding. In XFEM analysis, both fracture energy and displacement can serve as criteria for damage evolution. However, for composite materials, defining fracture energy in each direction is challenging. Therefore, displacement is used as a substitute, and the displacement at failure for damage evolution was set to 2 mm.

A minimum burst pressure of 117 MPa was applied to observe the extension of cracks in the cylinder winding under high-pressure loading conditions. The comparison of crack extension states before and after aging is presented in Figure 18. STATUSXFEM represents the crack extension status, with numerical values ranging from 0 to 1. A value of 1 indicates complete cracking within the element, 0 signifies no cracks within the element, and other values represent incomplete cracking within the element.

From Figure 18, it can be observed that after material aging, the crack becomes more sensitive, with an increased propagation length and a lower initiation load. After aging for 50 days at 70 °C, the crack initiates expansion at a load of 78.98 MPa, which is 10% lower than the non-aged condition. When loaded to 117 MPa, the right end of the crack essentially penetrates through the fourth layer of the helical winding, resulting in a transverse total length of approximately 3.684 mm, which is 0.614 mm longer than the non-aged condition.

Figure 19 displays the maximum principal stress contour plots before and after material aging under an internal bursting pressure of 117 MPa and at the initiation of crack propagation. After 50 days of aging at 70 °C, under the same pressure conditions, the crack propagation length increases, and the maximum principal stress at the crack tip rises to 4.943 GPa, representing a 3.87% increase compared to the non-aged state.

As shown in Figure 20, the relationship curve between the maximum principal stress and the layers is presented before and after UV aging at 117 MPa. In comparison to the case without a crack, stress concentration occurs at the crack tip in the second layer, leading to a noticeable increase in the maximum principal stress in the second layer. In the presence of a crack, the maximum principal stress in the second layer increases after UV aging. The third layer, not located at the crack tip, exhibits minimal difference before and after aging. In the fourth layer, the varying extent of crack propagation results in a noticeable difference in stress concentration at the crack tip. The maximum principal stresses in the other layers show minimal variation.

UV aging leads to a reduction in the tensile strength and modulus of CFRP, a decrease in interlayer bonding, and a decline in the maximum stress that fibers and matrices can withstand. Consequently, the material layers experience premature failure at the same stress level after aging. UV aging diminishes the overall load-bearing capacity and crack resistance of the hydrogen storage cylinder, posing greater safety risks during service.

## 5. Conclusions

This study investigates the impact of different durations of UV irradiation at various temperatures on the Mode I fracture toughness, Mode II fracture toughness, and mixed-mode fracture toughness of CFRP laminates. Additionally, leveraging the post-UV aging performance data, an XFEM-based analysis is conducted on the crack propagation behavior in Type IV hydrogen storage cylinders with pre-existing cracks. The conclusions are as follows:(1)UV aging has a significant impact on both Mode I and Mode II fracture toughness. With increasing aging duration, Mode I fracture toughness increases, while Mode II fracture toughness decreases. After 50 days of aging at 70 °C, compared to the non-aged specimens, $G_{IC}$ increases by 22.1%, and $G_{IIC}$ decreases by 18.2%. The experiments indicate that as the aging time extends, the interlayer bonding strength of the CFRP laminate weakens, leading to a decrease in Mode II fracture toughness. However, the number of fiber bridging between layers gradually increases, significantly enhancing Mode I fracture toughness.(2)UV aging also has a noticeable influence on mixed-mode interlaminar fracture toughness. With the increasing aging duration, mixed-mode fracture toughness exhibits a trend of initial increase followed by a subsequent decrease. Under the conditions of 70 °C aging, as the aging time increases, $G_{C}$ reaches its peak, showing a maximum increase of 7.2%. Even after 50 days of aging, there is still a 2% improvement. The experiments reveal that in the early stages of UV aging, the post-curing of the resin enhances the material’s toughness. However, in the later stages, the damaging effect of UV radiation surpasses the beneficial effects of post-curing, leading to a significant reduction in interlayer bonding strength.(3)Due to the actual operating conditions of high-pressure hydrogen storage cylinders, cracks perpendicular to the length direction of the vessel are more likely to cause damage than parallel cracks. Transverse damage may result in various failure modes, including fiber fracture, matrix deformation and cracking, fiber-matrix separation (fiber debonding), and fiber pull-out. Therefore, the analysis of crack propagation in Type IV hydrogen storage cylinders cannot be simplified based solely on fracture toughness. The effect of UV aging on the cross-layer expansion of cracks in Type IV hydrogen storage cylinders was simulated and analyzed by XFEM. The results indicate that the cracks in the aged composite winding layer are more sensitive, exhibiting lower initiation loads and longer crack propagation lengths under the same load. UV aging reduces the tensile strength and modulus of CFRP, causing a decline in the maximum stress fibers and matrices can withstand. At the same stress level, the material layers enter failure states earlier after aging. UV aging diminishes the overall load-bearing capacity and crack resistance of hydrogen storage cylinders, posing greater safety risks during service.

## Figures and Tables

**Figure 1 materials-17-00846-f001:**
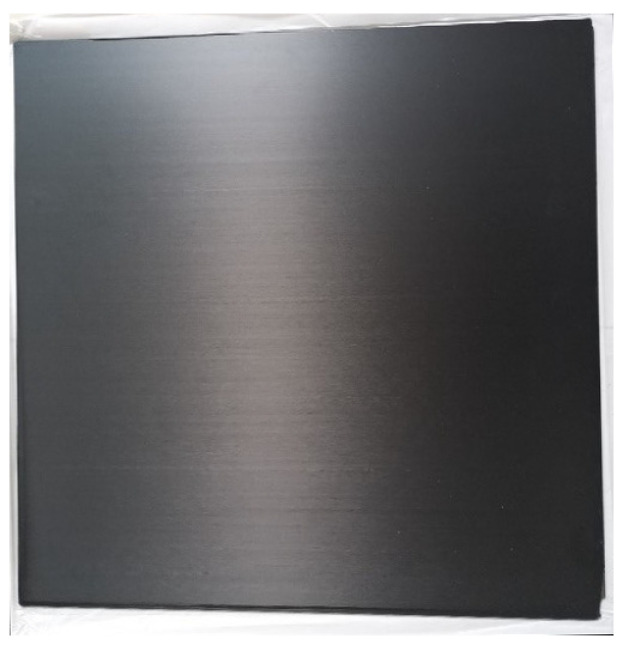
CFRP laminate.

**Figure 2 materials-17-00846-f002:**
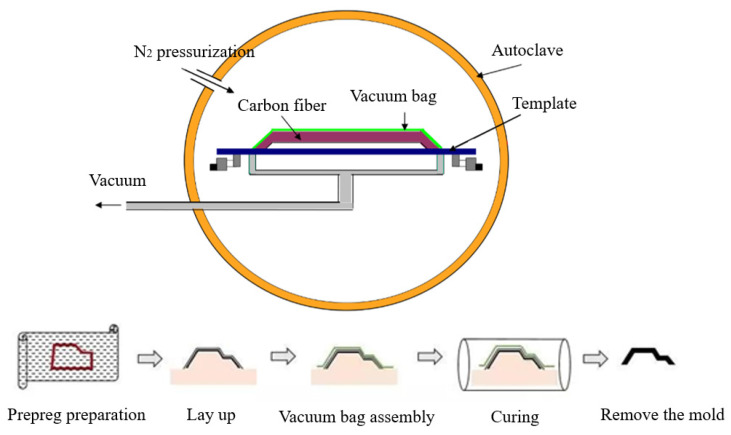
Flow chart of autoclave molding process.

**Figure 3 materials-17-00846-f003:**
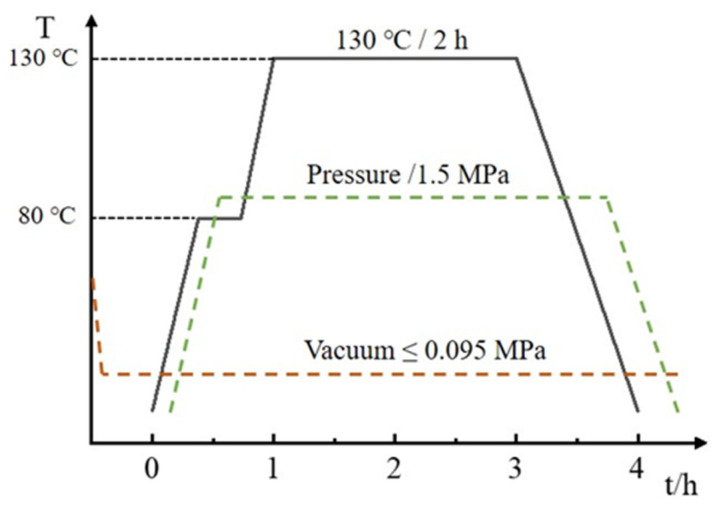
Curing process parameters.

**Figure 4 materials-17-00846-f004:**
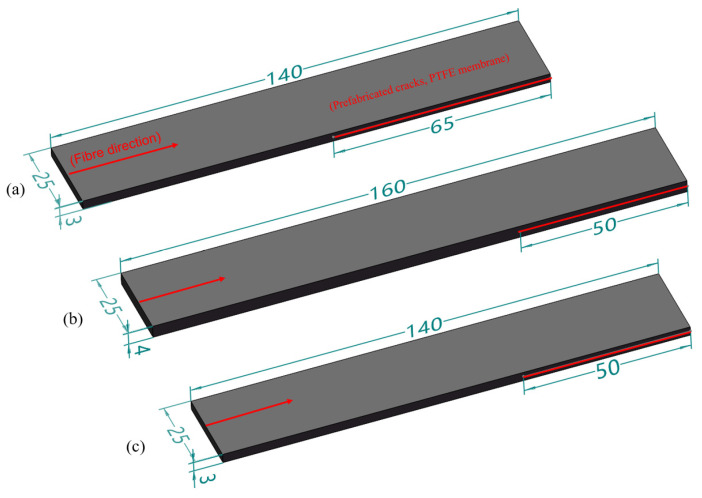
Specimen model and dimensions (unit: mm). (**a**) Type I fracture specimens; (**b**) Type II fracture specimens; (**c**) mixed fracture specimens.

**Figure 5 materials-17-00846-f005:**
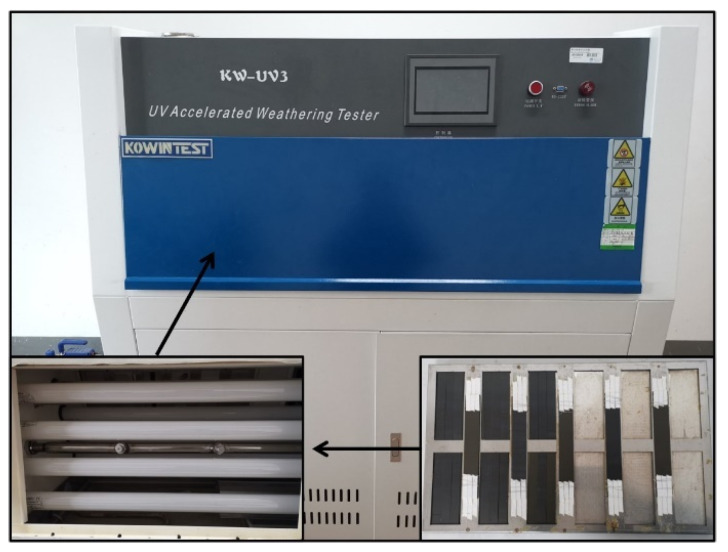
UV exposure chamber.

**Figure 6 materials-17-00846-f006:**
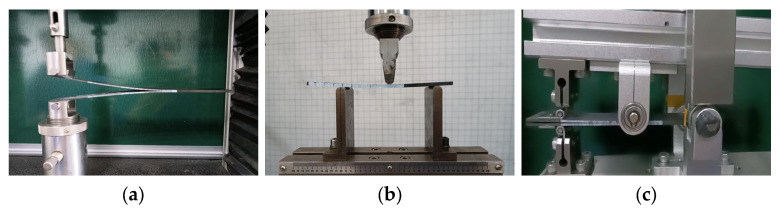
Fracture toughness test apparatus. (**a**) Double Cantilever Beam (DCB) Test apparatus; (**b**) End Notched Flexure (ENF) Test apparatus; (**c**) Mixed-Mode Bending Test apparatus.

**Figure 7 materials-17-00846-f007:**
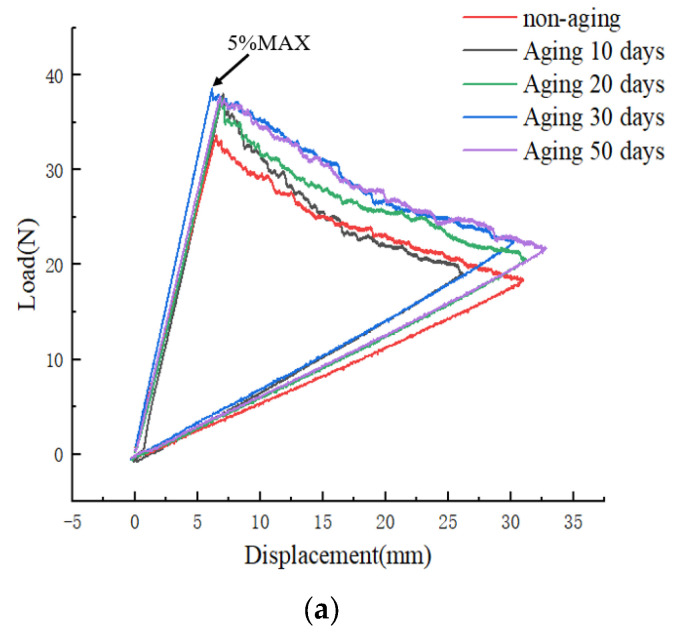
(**a**) Load–displacement curves at different aging times; (**b**) R-curves at different aging times.

**Figure 8 materials-17-00846-f008:**
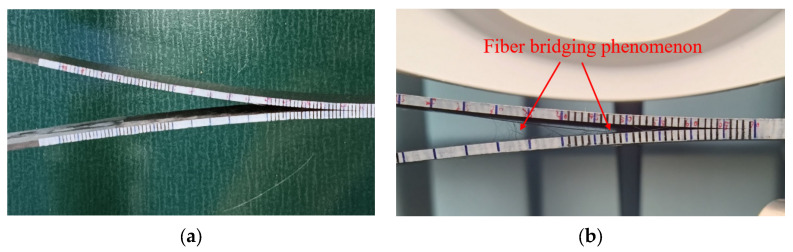
(**a**) Fiber bridging phenomenon in non-aging specimens; (**b**)Fiber bridging phenomenon at 70 °C aging for 10 days.

**Figure 9 materials-17-00846-f009:**
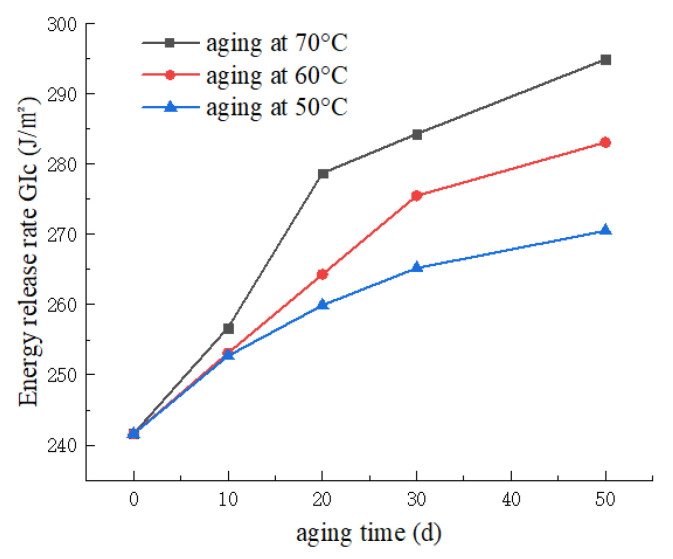
Variation curve of energy release rate with aging time at different temperatures.

**Figure 10 materials-17-00846-f010:**
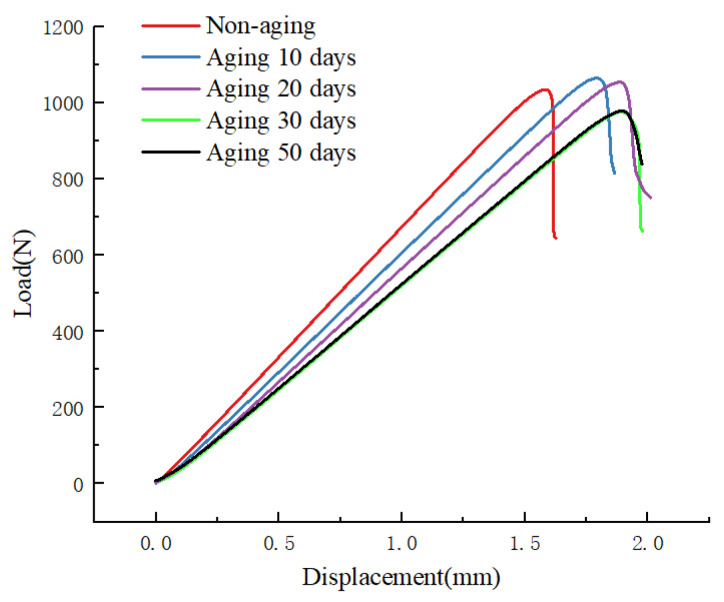
PC loading load–displacement diagram for different aging days.

**Figure 11 materials-17-00846-f011:**
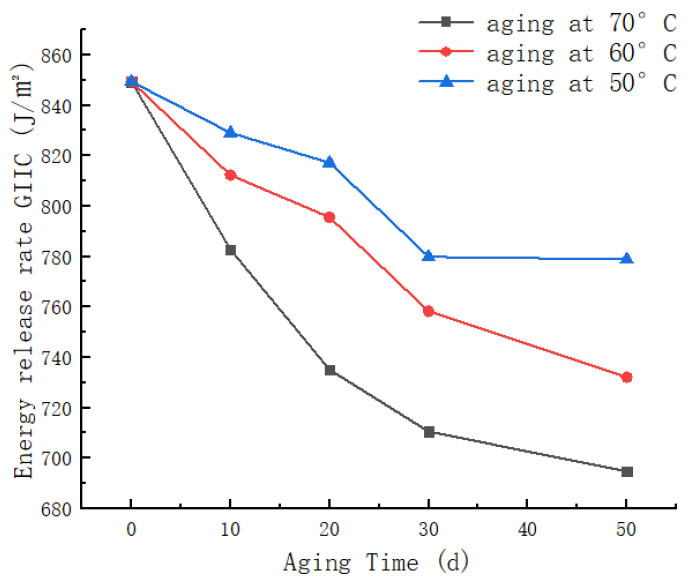
Variation curves of type II energy release rate with aging time at different temperatures.

**Figure 12 materials-17-00846-f012:**
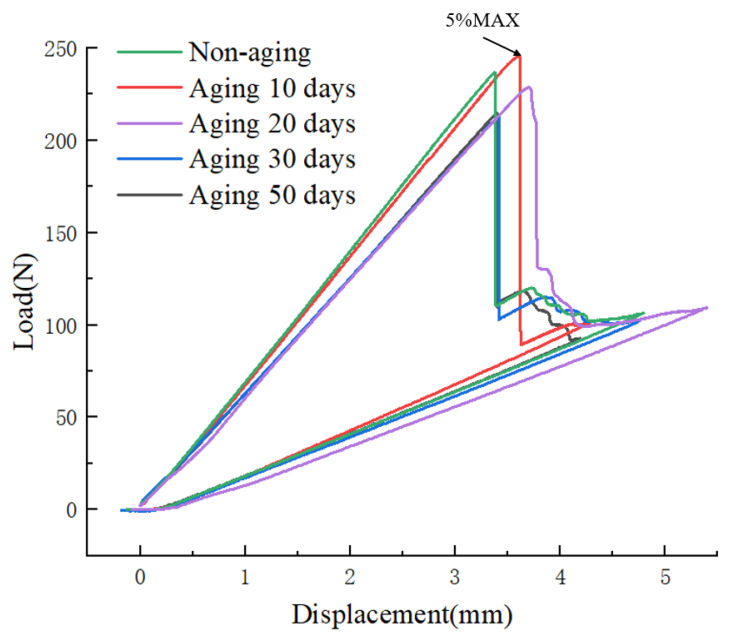
Variation of load–displacement curves under different aging days.

**Figure 13 materials-17-00846-f013:**
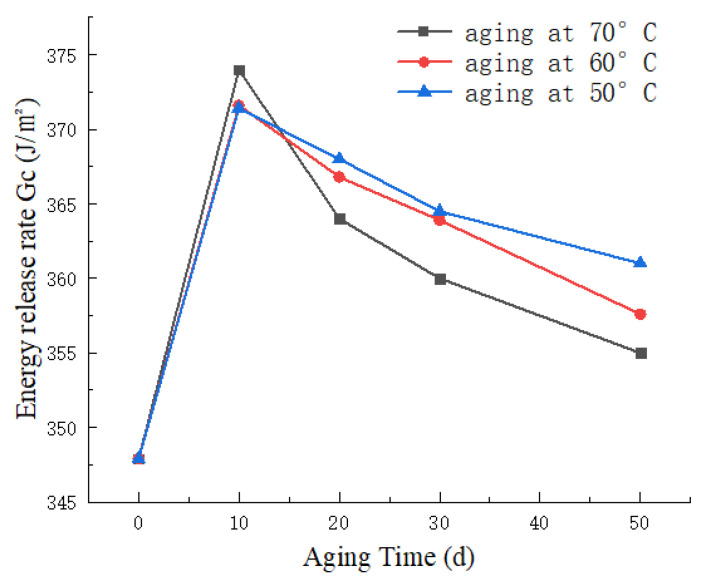
Variation curve of mixed fracture toughness with aging time at different temperatures.

**Figure 14 materials-17-00846-f014:**
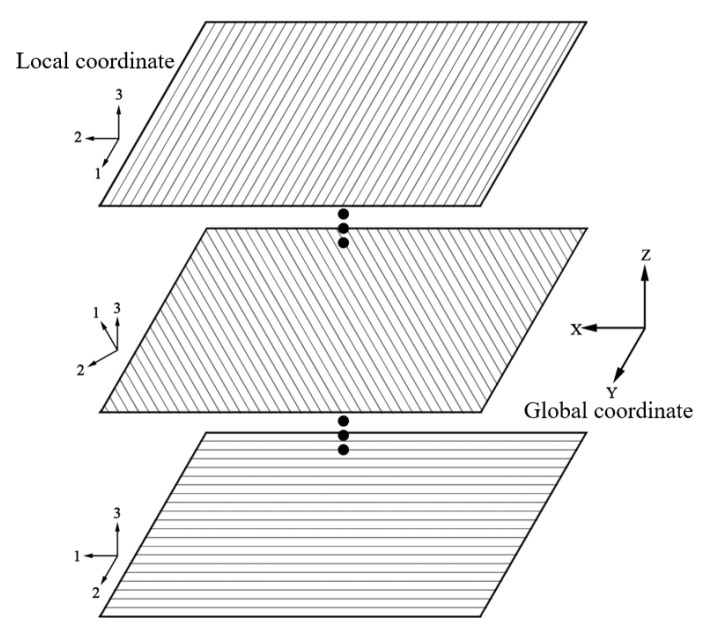
Schematic of the ply modeling method.

**Figure 15 materials-17-00846-f015:**
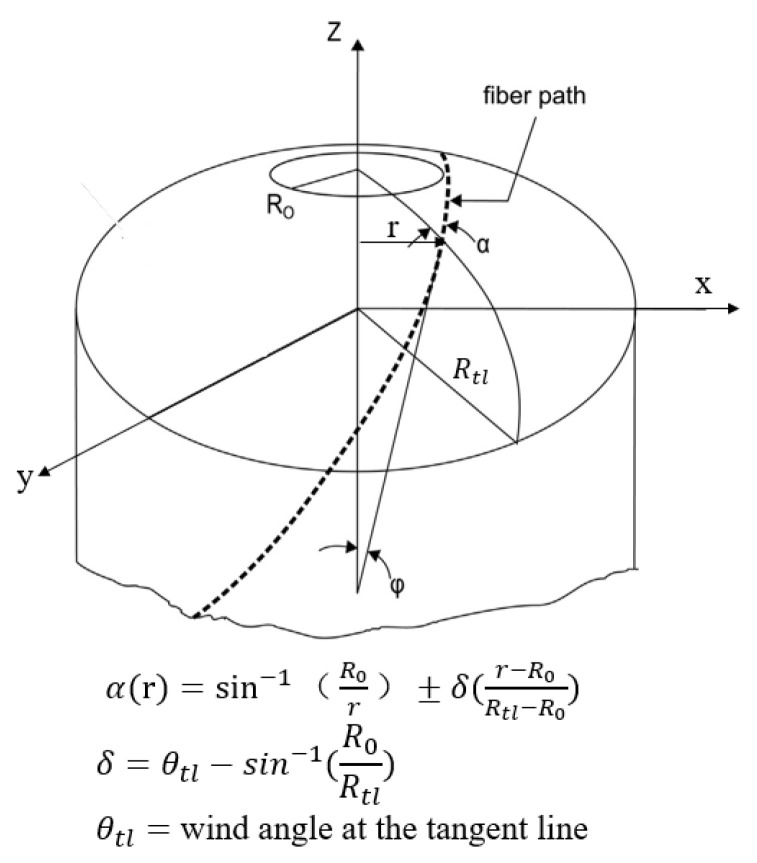
Variation of helical angle in the dome section.

**Figure 16 materials-17-00846-f016:**
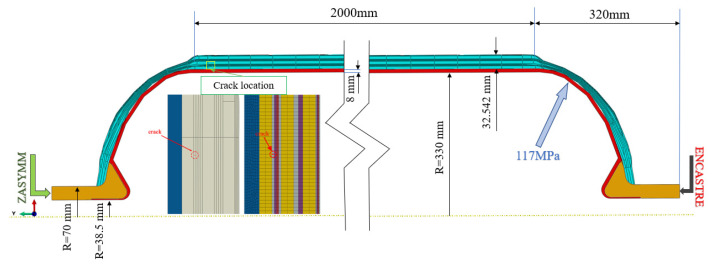
Type IV hydrogen storage cylinder dimensions.

**Figure 17 materials-17-00846-f017:**
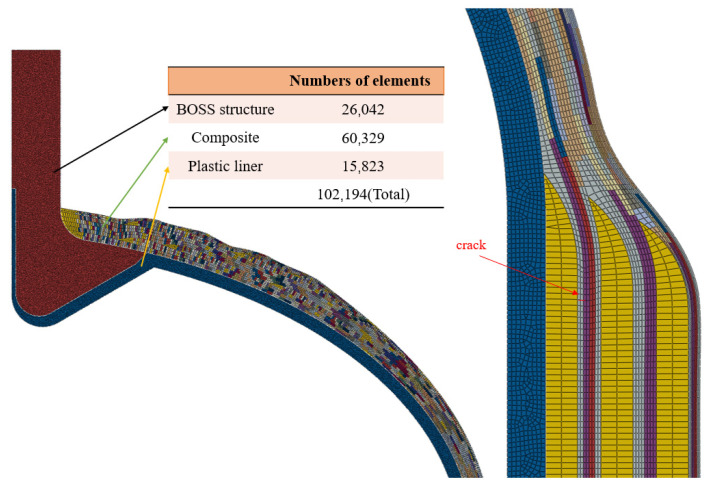
Schematic representation of the material with the color WCM element.

**Figure 18 materials-17-00846-f018:**
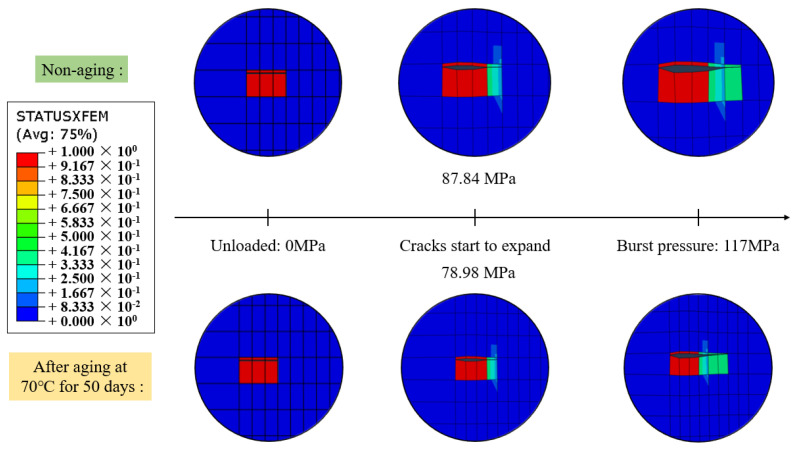
Comparison of crack extension state before and after aging.

**Figure 19 materials-17-00846-f019:**
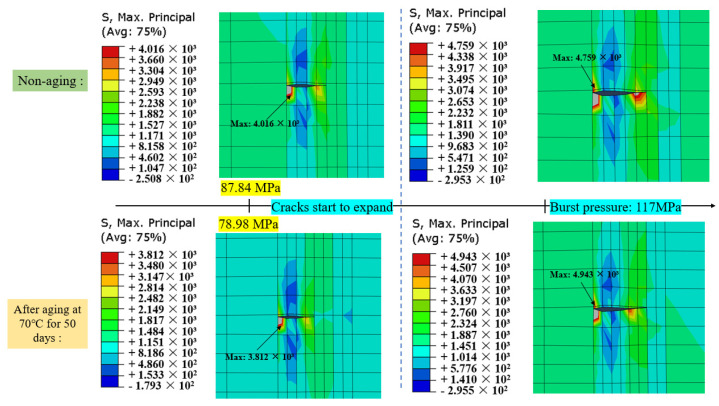
Comparison of maximum principal stresses before and after aging.

**Figure 20 materials-17-00846-f020:**
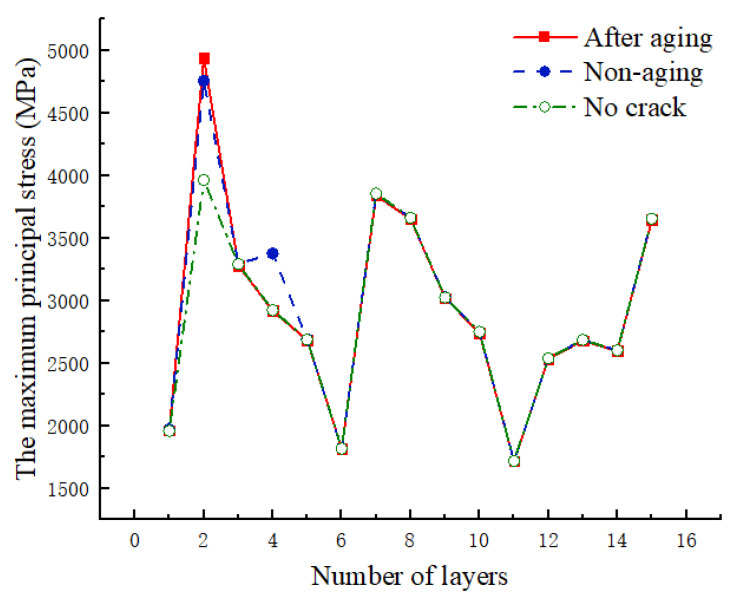
Variation of the maximum principal stress value with the number of layers laid before and after aging.

**Table 1 materials-17-00846-t001:** T700S carbon fiber main parameters [24].

Fiber Type	Filaments	Tensile Strength (MPa)	Young’s Modulus (GPa)	Elongation (%)	Density (g/cm^3^)
T700S	12 k	4900	230	2.1	1.8

**Table 2 materials-17-00846-t002:** Resin system’s main parameters.

Epoxy Resin	Appearance	Viscosity (Pa.s)	Epoxy Value (eq/100 g)	Volatile Components (%)
Y04	Light yellow viscous liquid	10–16	0.48–0.54	≤2

**Table 3 materials-17-00846-t003:** UV aging cycle method.

Aging Cycle	Aging Phase	Wavelength (nm)	UV Irradiance (W/m^2^·nm)	Blackboard Temperatures (°C)	
T1	8 h dry 4 h condensation	340	1 ± 0.02 0.00	70 ± 3 50 ± 3	
T2	8 h dry 4 h condensation	340	1 ± 0.02 0.00	60 ± 3 50 ± 3	
T3	8 h dry 4 h condensation	340	1 ± 0.02 0.00	50 ± 3 50 ± 3	

**Table 5 materials-17-00846-t005:** Mechanical properties of AL6061-T6 and PA6.

	AL6061-T6	PA6
E, MPa	69,000	358.11
ν	0.33	0.4
Tensile strength, MPa	368	~
Yield Strength, MPa	241	18

**Table 6 materials-17-00846-t006:** Fiber-wound layer layup parameters.

NO.	Layer Type	Wind Angle (°)	Thickness (mm)	Band Width (mm)
1	Hoop	90	6.754	20
2	Helical	12	1.228	20
3	Helical	18	1.228	20
4	Helical	23	1.228	20
5	Helical	36	1.228	20
6	Hoop	90	6.754	20
7	Helical	12	1.228	20
8	Helical	12	1.228	20
9	Helical	18	1.228	20
10	Helical	30	1.228	20
11	Hoop	90	6.754	20
12	Helical	36	0.614	20
13	Helical	30	0.614	20
14	Helical	23	0.614	20
15	Helical	12	0.614	20

## Data Availability

Data are contained within the article.

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
