# Peer review of "Fracture Performance Study of Carbon-Fiber-Reinforced Resin Matrix Composite Winding Layers under UV Aging Effect"

_materials, 2024, doi:10.3390/ma17040846_

Round 1
Reviewer 1 Report
Comments and Suggestions for Authors
The study examines the impact of different durations of ultraviolet (UV) irradiation at various temperatures on the fracture toughness in I-mode, II-mode, and mixed-mode for Carbon Fibre Reinforced Polymer (CFRP) laminates.
· In the abstract you stated that “The results indicate that with an increase in UV aging duration, the I-mode fracture toughness of the material increases” and “Post-aging, the material exhibits increased sensitivity to cracks….. and reduced starting loads for crack extension”. These two results are contradictory. Explain.
· The introduction section is too poor and you need to discuss previously published works in this area (UV aging, XFEM, Fracture, etc.).
· What is the novelty of the work?
· You shall explain why you used the XFEM technique instead of the other ones such as cohesive element modeling as you did DCB and ENF.
· The section “2. Experiment” is written like a short technical note. Revise and rewrite this section.
· Add a section regarding the UV-aging mechanism and how XFEM works.
· In Figure 1, what are a, b, and c??
· There are no details regarding the manufacturing process.
· There are no details regarding the materials supplier.
· Page 2, line 67: The standards mentioned here (ASTM D5528, ASTM D7905, and ASTM D6671) shall be referred to properly.
· The manuscript is not well-written and organized.
· Page 2, line 77: “a UV accelerated aging chamber.”. Name that chamber brand, manufacturer, etc. and add more details.
· Page 3, line 78: “Three different blackboard temperatures (70°C, 60°C, 50°C)”. How did you choose those temperatures? Did you consider the glass transition temperature of the resin?
· What standard test method did you employ to perform the UV-aging process?
· Add more details regarding performing the DCB, ENF and mixed mode tests.
· The quality of the figures is not acceptable.
· In Figure 13, what is the unit of the dimensions?
· In Figure 16, revise the wording error.
· Do you think it would be possible to use your results to predict the long-term life of composite materials exposed to UV? How? Something like what was discussed in the below works regarding the life prediction of aged composite materials in a hygrothermal environment. Discuss this type of work and the feasibility of life prediction in the manuscript.
“Multi-scale modelling and life prediction of aged composite materials in salt water”
“The Aging Behavior and Life Prediction of CFRP Rods under a Hygrothermal Environment”
“Indentation characterization of glass/epoxy and carbon/epoxy composite samples aged in artificial salt water at elevated temperature”
Author Response
Thank you for your comments. We have tried our best to improve the manuscript and have made a number of revisions, including important changes to the "Abstract", "Introduction" and "Experiments", and the images of the article have been redesigned. Please see the attached and updated manuscript for details.
We would like to express our sincere gratitude to the editors/reviewers for their hard work and hope that the revisions will be recognized.
Thank you again for your comments and suggestions.

Reviewer 2 Report
Comments and Suggestions for Authors
This paper provides different methods to study the fracture toughness of Carbon Fiber Reinforced Polymer (CFRP) laminates. UV aging and life prediction were conducted. The paper is of interest for materials journal but needs to be improved:
1- Initiate the abstract with the importance of fracture study in CFRPs. The novelty should be also provided later in the abstract, final sentence should be devoted to the contribution of the study to future works.
2- Introduction is really weak. You need to discuss the previous works and the details of their results related to your work. It should be restructured.
3- what is the novelty of your work. provide in last paragraph of intro please.
4- Needs reference for this sentence from materials journal:
The degradation of fibers or matrices may reduce the ability of the composite material 38 to effectively transmit loads between these components, resulting in changes in mechanical performance.
5- provide material properties in the related section
6- provide nomenclature for your paper
7- in figure 8 30 days and 50 days aging have similar trend. What does it show?
8- figure 11 should be in better quality and larger. Provide all figures in the middle of the page
9- what are the boundary conditions?
10- did you conduct convergence study for numerical simulation?
11- Provide the importance of CFRP materials in different applications
https://doi.org/10.1016/j.addma.2020.101728
12- figure 16 initial state is not written correctly in paper
13- figure 17 enlarge the bar chart and numbers could be seen easily
14- conclusion provide bullet points with most important findings
Comments on the Quality of English Language
Double check please.
Author Response

(The authors gave the same response as above.)

Round 2
Reviewer 1 Report
Comments and Suggestions for Authors
The authors have addressed the comments and questions in the new revision comprehensively. There are still some minor points that need to be considered.
It would be better not to use words like "our" and "we", etc. in a scientific work.
Add a reference to Table 1.
I think you can connect the great work you have done in this paper to the works mentioned in my previous comment regarding long-term life prediction. It is worth it to add a paragraph in the introduction to show that it is possible to use the Arrhenius formula to predict the life of composites in the investigated environment.
Author Response
Thank you very much for your comments, we have made changes to the article. Please see the attachment.

Reviewer 2 Report
Comments and Suggestions for Authors
Well done
Comments on the Quality of English Languagecheck again
Author Response
Thank you very much for your comments, we have made changes to the article.